# A Model for Diagnosing Autism Patients Using Spatial and Statistical Measures Using rs-fMRI and sMRI by Adopting Graphical Neural Networks

**DOI:** 10.3390/diagnostics13061143

**Published:** 2023-03-16

**Authors:** Kiruthigha Manikantan, Suresh Jaganathan

**Affiliations:** Department of Computer Science and Engineering, Sri Sivasubramaniya Nadar College of Engineering, Chennai 603110, India

**Keywords:** autism spectrum disorder, deep learning, graph convolution networks, sMRI, rs-fMRI

## Abstract

This article proposes a model to diagnose autism patients using graphical neural networks. A graphical neural network relates the subjects (nodes) using the features (edges). In our model, radiomic features obtained from sMRI are used as edges, and spatial-temporal data obtained through rs-fMRI are used as nodes. The similarity between first-order and texture features from the sMRI data of subjects are derived using radiomics to construct the edges of a graph. The features from brain summaries are assembled and learned using 3DCNN to represent the features of each node of the graph. Using the structural similarities of the brain rather than phenotypic data or graph kernel functions provides better accuracy. The proposed model was applied to a standard dataset, ABIDE, and it was shown that the classification results improved with the use of both spatial (sMRI) and statistical measures (brain summaries of rs-fMRI) instead of using only medical images.

## 1. Introduction

Autism spectrum disorder (ASD) is a neuro-developmental disability that affects the social and behavioural abilities of an individual. People with ASD find it challenging to interact with society and accept changes in their routine work [1,2]. The diagnosis of autism includes observations made by (1) paediatricians, (2) child psychologists, (3) neurologists, and (4) therapists. To analyse the behavioural pattern of a child, visits to a psychologist and paediatrician is needed, which can take from months to years. Some distinguishable findings of autism-affected brains are (i) an increased brain volume, (ii) reduced functional connectivity in the temporal lobe and visual cortex, and (iii) more folds in the left parietal, temporal and frontal lobes, areas that affect social behaviour and language production. Magnetic resonance imaging (MRI), structural MRI (sMRI), functional MRI (fMRI) and resting state fMRI (rs-fMRI) help with the early and accurate diagnosis of autism. sMRI data provide structural information about the brain, such as its size, volume, convolutions, grey matter density and white matter distribution. fMRI and rs-fMRI provide information about functional dependencies in brain regions, which can aid in the identification of a child’s social and behavioural activities [3,4]. The brain’s major characteristics can be determined by means of sMRI and rs-fMRI.

Radiomics is used in medical imaging to augment the available data by applying a mathematical analysis to quantitative features and extracting pixel correlations and the distribution of signal intensities in brain regions (voxels). The features obtained through radiomics are (i) shape and size-based, (ii) image density histograms, (iii) correlations between voxels, (iv) texture from filtered images, and (v) form factor features [5,6,7].

Given a graph G(V,E) as the input, the graph convolution network (GCN) takes an input as a matrix X(N×F), where *N* is the number of nodes and *F* is the features of a node, and an adjacency matrix A(N×N). A degree matrix *D* is a diagonal matrix that stores the sum of rows of an adjacency matrix *A* in the diagonal $D_{i}=\sum j\cdot A_{ij}$. Each node in the graph has a *d* dimensional feature vector, so the entire feature matrix can be described as $X\in R_{n\times d}$. All nodes in the graph belong to one of the *c* classes specified in the *c* dimensional vector $y_{i}\in 0,\;1,\;\ldots ,c$.

A GCN [8,9] is similar to a neural network learning new features of each node over multiple layers ( refer Figure 1). The initial node representation is $H_{0}=X$, the input to the GCN. In a *k*-layer GCN, the hidden layer representation is averaged with its neighbours at the beginning of each layer. A three-step process of nonlinear propagation, linear transformation and nonlinear activation occurs in each hidden layer.

Feature propagation involves averaging the feature of each node $v_{i}$ at the beginning of layer $H_{i}$ with its local neighbourhood, as shown in Equation (Equation 1).
(1)$$H_{i}=\frac{\sum_{j=1}^{n}A_{ij}}{\sqrt{\left(d_{i}+1\right)\left(d_{j}+1\right)}}$$The normalised matrix representation is specified in Equation (Equation 2)
(2)$$H_{i}=\frac{1}{\sqrt{\bar{D}}}\cdot \bar{A}\frac{1}{\sqrt{\bar{D}}}H_{i-1}\cdot W_{i-1}$$
where $\bar{A}=A+I_{N}$ and $\bar{D}=\sum j\bar{A_{ij}}$ is the Degree matrix of *A*.Feature and nonlinear transformations associate each layer with a learned weight matrix $W_{i}$, and hidden layer transformations are transformed linearly. A feature representation of layer $H_{k}$ is generated after applying ReLU. This step is followed by feature propagation for the next layer. Equation (Equation 3) provides the criteria for updating the kth layer [10].
(3)$$H_{k}\leftarrow ReLU\left(H_{k},W_{k}\right)$$The classification of a node in a kth layer GCN is obtained using Equation (Equation 4).
(4)$$Y_{GCN}=Softmax\left(H_{k}\right)$$

Analyses using brain summary methods typically involve the extraction of statistical features from the rs-fMRI data. These features can be used to quantify the strength and direction of functional connectivity between different brain regions to characterize the magnitude and consistency of the rs-fMRI signal within a given brain region. These statistical features can provide information on the underlying physiological processes and can be used to identify abnormalities in brain function. Brain summary measures are used to reduce the 4D time series data using statistical measures into a single number per voxel. The summary measures chosen are listed below:First, the Regional Homogeneity (ReHo) uses Kendall’s coefficient of concordance to calculate the similarity between the time series of a voxel and its neighbour.Second, the Amplitude of Low-Frequency Fluctuations (ALFF) uses the amplitude of each region of interest (ROI) in the frequency band of 0.1 to 0.08 Hz. It shows the spontaneous neuro behaviour in rs-fMRI data. The power spectrum is obtained from the time series signal by a fast Fourier transform. The averaged square root value of each voxel in the power spectrum across the frequency range is considered the ALFF value.Third, the Fractional Amplitude of Low-Frequency Fluctuations (fALFF) finds the power ratio in the frequency band relative to the full frequency band of 0 to 0.25 Hz.Fourth, the Degree Centrality ($DC_{bin},DC_{weighted}$) measures the edge count of a region. It specifies the number of regions to which the current node is related. This measure proves higher associations in cortical regions.Fifth, the Eigenvector Centrality ($EC_{bin},EC_{weighted}$) provides the centrality of a node based on the number of edges connecting central nodes; i.e., a node is important if it is connected to more important nodes.Sixth, the Local Functional Connectivity Density (LFCD) calculates the local functional connection of a voxel within a ROI and the functional connections of voxels between hemispheres.Seventh, the Voxel-Mirrored Homotopic Connectivity (VMHC) calculates the voxel-wise connectivity between symmetrical brain hemispheres. ROIs that show high connectivity between the left hemisphere, and its right mirrored counterpart have high VHMC values, as shown in Figure 2.Eighth, dual regression uses the independent component group maps obtained from the independent component analysis (ICA) as network templates. A spatial regressor uses the spatial map to identify the time series associated with voxels on the corresponding map, followed by a temporal regressor, which uses the time series to fetch the complete set of voxels activated in that time series. The result of these steps is a subject-specific spatial map based on the original spatial map.

The motivation behind the proposed work was to use both structural features and functional connectivity to contribute to a study of the brain. The summary measures provide the most essential statistical features of the brain. More accurate results could be obtained by fusing more summaries instead of obtaining images from a single summary measure.

The highlights of our proposed model are as follows:Radiomic features obtained from sMRI using the Destrieux atlas were used to identify similarities between individuals (nodes) to construct an edge for the graph. These features can capture information about various aspects of the image, such as the texture, shape, size, intensity, and spatial relationship.We extracted 48,544 radiomic features passed to stacked autoencoders for a dimensionality reduction. We obtained a similarity index between the 150 features of each node using an improved sqrt cosine similarity function to draw an edge between the nodes.Brain summaries provide spatial features passed to the Multichannel CNN to obtain the feature vector for each node. We obtained 3D volume images for each brain summary derivative and combined them on the fourth (channel) axis. The multichannel 3D CNN produces a feature vector with a size of 1024.Population Graphs were constructed with each subject as a node and edges based on the similarities identified in the brain’s structure. The consideration of structural similarities reduced the heterogeneity of the ABIDE dataset, as MRI data were collected from 16 locations.Graph convolution networks were used to produce complete graphs, which helped to classify nodes as ASD or Typical Controls (TC).

The rest of the paper is organized as follows: Section 2 discusses the work related to the GCN and deep learning methods in terms of classifying ASD from TC. Section 3 elaborates on the proposed method, Section 4 explores the experimental results, Section 5 details the research discussion, and finally, Section 6 concludes the paper with possible future work.

## 2. Related Work

Brain scans and deep learning algorithms can be used to classify ASD in two different ways: the graphical approach and the nongraphical method. This section details research on the classification of ASD using these two methods. We also discuss how radiomics may be used to extract features through various brain imaging techniques.

### 2.1. Nongraph Deep Learning Methods

Deep neural networks such as convolutional neural networks (CNN), autoencoders (AE), recurrent neural networks (RNN), and the long short-term memory (LSTM) networks CNN and AE are frequently used for the extraction of features from images. Convolutional neural networks (CNNs) are commonly used for image-processing tasks such as image classification, object detection, and image segmentation. The use of CNNs for image processing has been highly successful due to their ability to learn and extract features from substantial amounts of data, making them well-suited for tasks where the data are highly structured and where deep learning has been proven to be a practical approach.

Ahammed et al. used a 2D CNN model to learn features from 3D fMRI images by increasing the number of convolution layers. They used 20 convolutional layers [11]. Increasing the number of layers allows the model to learn more complex representations of the input data and capture higher-level features. They considered only the data from the NYU ABIDE1 dataset. Topographical heterogeneity was not considered in their study. Increasing the number of convolution layers might allow more features to be lewarned, but the drawback is overfitting and an increased execution time. This could be overcome using the model proposed by Reem Haweel et al. They used K-means clustering to cluster BOLD signals in fMRI data and the continuous wavelet transform to get a detailed representation of BOLD signals [12]. The multichannel 2D CNN was used for classification. The wavelet transform used in the model can be sensitive to the choice of parameters, such as the wavelet type and decomposition level. If the decomposition level is not chosen properly, wavelet transformation might lead to overfitting, which would affect the performance of the model. Furthermore, interpreting the wavelet coefficients can be challenging and may not always align with the underlying biological processes being studied. Li et al. discussed a two-channel 3D convolution network where the two channels are the mean and standard deviation of the temporal fMRI data using a sliding window [13]. Leming et al. derived a structural similarity metric with grey matter volume data from structural MRI images. Their model is specific to the methods used to derive the brain connectome [14]. Generalization needs to be performed.

Autoencoders can extract features from an image by compressing the image data into a lower-dimensional representation, known as the encoding, and then reconstructing the image from the encoding. The encoding produced by the autoencoder typically includes the most vital information or “features” of the image, while less relevant information is discarded. ASD-DiagNet [15] is a hybrid learning approach that combines the strengths of deep learning and conventional machine learning to detect autism using fMRI data. This approach is based on the idea that deep learning can capture the complex patterns in fMRI data, while conventional machine learning methods can provide more interpretable solutions.

Sewani H. et al. proposed an autoencoder-based deep learning model followed by a sequence of two 1D CNNs [16]. They used only Pearson’s correlation matrix to extract features from fMRI. No specific atlases or statistical measures were used to obtain regions of interest. Compared to the machine learning methods specified in [15] CNNs are scalable and can be generalized. Liaqat Ali et al. proposed an ensembling method for combining the features obtained from different modalities. The fusing of inputs from different modalities can improve the number of features learned compared with the use of a single type of input. They combined two methods: fusion blending and voting [17]. First, all features from various modalities were fused without ranking or extracting the importance of the features selected.

### 2.2. Graph-Based Deep Learning Methods

GCNs aim to learn node-level representations by aggregating information from the surrounding nodes and edges. The EigenGCN [18], proposed by Yao Ma et al., uses eigen-based decomposition pooling, which summarises the node information to generate a graphical structure. Several pooling layers introduced between the GCN transfer the graph into a coarsened version where the number of nodes is reduced. Due to the pooling layers, crucial information about the nodes is lost. Instead of node-level learning, multiple views of a graph can be aggregated to form a super node. The MVS-GCN [19] is a machine learning method proposed for diagnosing ASD. The approach combines the information from multiple sources or “views” of data and uses a GCN to model the relationships between data points. The method also includes a prior brain structure learning component, which means that it incorporates prior knowledge of the brain structure into the model to improve its performance. Xuegang Song et al. proposed a method for predicting disease-induced deterioration using a graph convolution network that incorporates similarity awareness and adaptive calibration [20]. The “similarity awareness” aspect of the model suggests that the network is designed to consider the similarities between different nodes in the graph, which can be used to improve its predictions. The “adaptive calibration” aspect of the model refers to the ability of the network to adjust its parameters based on the specific characteristics of the data it is processing, which can lead to an improved accuracy level. Combining these two factors will improve the model’s prediction of disease-induced deterioration. Both methods described above use fMRI data only.

The Multi-Scale Enhanced Graph Convolutional Network (MESGCN) [21] uses graph convolutional networks (GCNs) for the detection of mild cognitive impairment (MCI). MCI is a condition in which a person experiences a decline in cognitive abilities but has not yet reached the stage of dementia. In the MESGCN method, the brain is represented as a graph, where each node represents a brain region, and the edges represent the connections between these regions. The GCN is then used to process this graph, where the node features are updated based on the features of the neighbouring nodes. The multiscale aspect of the MESGCN refers to the use of multiple scales in the graph representation to capture distinct levels of information in the brain. Considering the brain as a graph, connections between nodes are identified based on the functional connectivity. Other modalities could be used to generate a better graph representation of brain images. Parisot et al. used a GCN network to create a population graph in which the functional connectivity matrix is fed to an autoencoder to obtain a set of features for the nodes and phenotypic data are used to find similarities between nodes to construct edges [22]. The phenotypic data in the ABIDE 1 dataset were collected from North American and Western European populations, so the results need to be more generalizable to other populations.

### 2.3. Radiomics

Radiomics can be used to study the features of brain images, such as the white matter integrity, cortical thickness, and lesion load, which can provide essential information about the underlying structure and function of the brain. This information can be used to track the progression of diseases like multiple sclerosis, Alzheimer’s disease, and other neurodegenerative conditions, helping to improve the accuracy of diagnosis and prognosis.

Yao X et al. used radiomics features to identify spontaneously abnormal brain activities that could be used as Parkinson’s disease biomarkers [23]. The main limitation of the work is that it did not include cerebellum or multimodal data. The cerebellum contributes to various cognitive and affective functions, and it has been shown to have extensive connections with the cerebral cortex and other brain regions. Wang L. et al. investigated the use of textural features derived from functional magnetic resonance imaging (fMRI) of the hippocampus to improve the diagnosis of Alzheimer’s disease (AD) and amnestic mild cognitive impairment (aMCI). The hippocampus is a region of the brain known to be affected by AD and aMCI. The authors aimed to use imaging data to extract textural features that reflect local activity in this region [24]. Only the hippocampus was considered in their work, and other statistical measures of the brain need to be addressed. Table 1 provides a summary of related works in the field of diagnosing autism.

## 3. Materials and Methods

In this section, we present an overview of the proposed model, followed by an explanation of the proposed model for the classification of a subject as having ASD or being a healthy control given sMRI and rs-fMRI data based on our graph convolution model.

### 3.1. Proposed Method

The workflow of the proposed model using a graph convolutional network for classifying ASD from TC is illustrated in Figure 3, which uses both the structural and functional connectivity of the brain. The morphological abnormalities observed in the brains of ASD patients have variations in grey and white matter volumes and the sizes of cerebral, cerebellar, and subcortical regions. The intrinsic functional connectivity between the regions of the brain was identified using rs-fMRI.

The sMRI image obtained from the dataset is passed to the Deistreiux atlas for ROI extraction (step 1). A set of quantitative features is extracted from the segmented regions of the sMRI image. These features are classified into different groups, such as intensity-based features, texture-based features, and shape-based features. The features collected through radiomics are ranked using Fischer’s score, and the top 2000 features are selected (step 2). They are passed to stacked autoencoders to extract the top 150 features for each subject (step 3). The *Sqrt-cosine* similarity [25] is calculated between the nodes to identify whether an edge can be drawn between them (step 4).

Brain summary methods are commonly used in rs-fMRI analyses to summarize the functional connectivity patterns within the brain [26]. These methods involve the extraction of summary measures from rs-fMRI that capture the connectivity between different brain regions, thus reducing the temporal component of the 4D rs-fMRI image (step 5). Each brain summary method produces a 3D volume of a rs-fMRI image, consisting of 19 volumes of 3D images. They are passed to the multichannel 3D CNN, which yields 1024 features for each subject (step 6). Then, the complete dataset can be represented as a graph, where each subject is a node with 1024 features, and the similarity measure can be used to determine the edges between other nodes (step 7). They are then passed to the GCN, which produces a complete graph that helps to classify an unidentified node (step 8).

### 3.2. Building the Adjacency Matrix

The edges between the nodes of the graph are identified using the adjacency matrix. Then, the radiomic features obtained from sMRI are used. Radiomic features are used to better understand the textural and volumetric features of brain regions. Py-Radiomics (refer Figure 4) is used to extract the radiomic features from the preprocessed sMRI images. The wavelet transform in Py-Radiomics uses the Daubechies wavelet filter as the default wavelet filter. The high-pass filter of the Daubechies wavelet filter is applied in 8 different directions, corresponding to the 8 taps or coefficients of the filter. Therefore, the wavelet transform at level 1 in Py-Radiomics produces a set of sub-bands, each corresponding to a different frequency band and spatial orientation. After the wavelet transform has been performed, the resulting wavelet coefficients are used as the input to a feature extractor. This calculates a large number of features from the wavelet coefficients, including first-order statistics and texture features.

The Destrieux atlas, also known as the Cortical Parcellation atlas, is used to capture the structural information, such as the cortical thickness, grey matter volume, and curvature, calculated from the MRI images. The Destrieux atlas selects 74 regions from the right hemisphere and 74 regions from the left hemisphere, giving a total of 148 regions of interest (ROI). We obtain 19 histogram-based features, 22 texture-based (GLCM) features, and wavelet-based features in 8 directions. Overall, we have 328 features i.e., (19+22)×8.

The wavelet features are obtained from all possible directions of Low (L) and High (H) pass filters. Table A1 provides the radiomic features extracted with the help of Py-radiomics [27], and Table A2 specifies the terms used. We obtain 328 features for each ROI; for each subject, there will be 48,544 features, i.e., (328×148). Fisher’s score is a supervised algorithm that is used to select features (refer to Algorithm 1 and Table A3), and it returns each variable’s rank based on its score in descending order.
**Algorithm 1:** Feature Ranking using Fischer’s Score
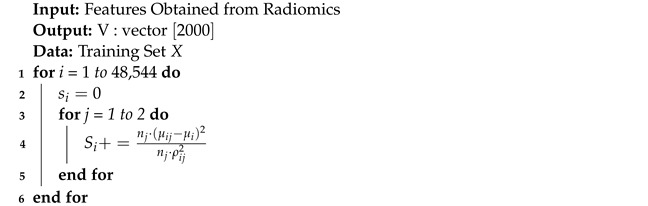


    Two thousand features are selected using Fisher’s score and passed to a stacked autoencoder (4 levels) for dimensionality reduction. Stacked autoencoders produce a latent space that is less sparse, more predictable, continuous [23,24] and yields 150 features. Algorithm 2 specifies the encoding process of the stacked autoencoder.
**Algorithm 2:** Stacked Autoencoder
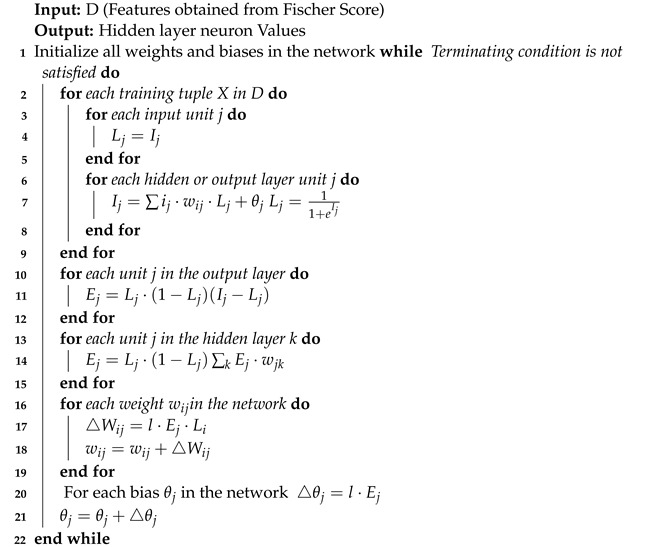


The level of similarity between nodes is identified using the *improved sqrt-cosine measurement*, as mentioned in Equation (Equation 5), and is based on the Hellinger distance. The main advantage of the Hellinger distance over the Euclidean or Manhattan distance sis that it is better suited for comparing probability distributions. The Euclidean and Manhattan distances are commonly used in traditional distance metrics and are useful for measuring distances between the vectors of numerical values.
(5)$$Sim\left(x,y\right)=\frac{\sum_{i=1}^{150}\sqrt{x_{i}\cdot y_{i}}}{\sqrt{\sum_{i=1}^{150}}x_{i}\cdot \sqrt{\sum_{i=1}^{150}}y_{i}}$$However, they are not well suited for measuring distances between probability distributions, because they do not take into account the structure of probability distributions, which is fundamentally different from that of vectors. The Hellinger distance, on the other hand, is specifically designed for comparing probability distributions and takes into account the structure of the distribution. It is defined as the square root of the sum of the squared differences between the square roots of the probabilities. This metric is useful for comparing distributions because it is bounded between 0 and 1, with 0 representing identical distributions and 1 representing completely different distributions. As it is similarity-based, it is associated with values between 0 and 1, which could be better associated with probability-based approaches. Thus, every node is connected to at least one other node, maximizing the sparsity. The threshold is set as the minimum of all maximum values in a row.

### 3.3. Feature Selection

rs-fMRI is used to identify the brain connectivity by calculating the blood oxygen levels, where the higher the oxygen level in a brain region is, the greater the activation of the region is [28]. We use brain summaries that are statistical derivatives to eliminate the temporal component of the MRI scan [29] and 3D CNN to extract features from them (refer Figure 5).

For each subject, the ten brain summary derivatives are calculated from the 4D fMRI data, resulting in a 3D volume brain image of 61×73×61, meaning that we have a stack of 61 slices of 61×73 2D images. The spatial characteristics of the brain image, such as the brain regions, volume, cortical folding, and white matter tracts, are restored. Each volume can be viewed as axial, sagittal, or coronal.

For the dual regression derivative, ten different volumes, 61×73×61 in size, (10 spatial group maps from the Independent Component Analysis (ICA)) enhance the contrast of the brain tissue and separate unknown signal sources into statistically independent components. Maps 1, 2 and 3 strongly correspond to the visual behavioural domain; Map 4 is the default mode network; Map 5 corresponds to the cerebellar region; Map 6 contains sensorimotor information; Map 7 includes auditory paradigms, Map 8 corresponds to cognition and Maps 9 and 10 are left–right mirrors, which correspond to the perception and language paradigm). In total, there are 19 3D brain volumes for each subject. A fourth dimension, the channel axis, concatenates different derivatives as a multichannel. Dual regression produces images, 61×61×73×10 in size (10 denotes the channel for each map). When the ALFF needs to be concatenated with a dual regression to form a new multichannel, we add a channel axis, so its dimensions become 61×61×73×1. Now the concatenation is performed on the channel axis, resulting in new multichannel data with dimensions of 61×61×73×11.

We build a 3D convolution model to obtain features from the multichannel brain volumes. A convolution block is characterized as a convolution layer with ReLU activation followed by batch normalization and max-pooling. The Kernel size is set as (3×3×3), and the pool size is (2×2×2). The second convolution block is repeated four times, starting with 32 filters and doubling at each block, so the last convolution block has 256 filters. The global average pooling layer generates channel descriptors by 3D pooling of all previous layers. The flattened vector is given as the input to a dense layer consisting of 1024 nodes. Thus, 1024 features are obtained for each node by using the multichannel 3D CNN.

### 3.4. Graph Convolution

The population graph is constructed as nodes with 1024 features each, and the edges are determined with their similarity measured by the radiomic features of the sMRI data. In the GCN-based model, as specified in Figure 6, features are used to characterize the nodes, while edges denote the associations between the nodes [30,31]. The graph convolutional network uses two hidden layers, with each node containing 16 units with dropouts of 0.005 and 0.4, respectively.

In contrast to similar work carried out previously, in nongraphical-based methods, the correlation coefficients obtained from the functional connectivity matrix are used. The ASD-Diagnet [15,16] uses functional connectivity, a measure of the temporal correlations between different brain regions. The method has been used extensively to study the neural basis of ASD. Correlation coefficients can be influenced by head motion, the signal-to-noise ratio, and individual differences in brain anatomy. Failure to account for these confounding factors may result in errors or a reduced classification accuracy. Multimodal data classification [17] is performed by ensembling the classification results of different modalities by either blending or voting. Different modalities are only ensembled after classification. Ensembling multiple modalities before classification can improve the accuracy compared to ensembling the classification results. The accuracy is improved because each modality can capture different aspects of the data, and by combining them, a more comprehensive representation of the data can be created.

In graph-based methods, s-GCN [22] creates a graph with phenotypic data (edges) and functional connectivity features from rs-fMRI data used as nodes. The use of phenotype data to construct edges might lead to a lack of heterogeneity, since phenotype data do not consider variations in topographical locations. The MVS-GCN [19] generates multiple view graphs from brain subnetworks, which may result in redundant feature learning. The heterogeneity of the brain is not considered when forming subnetworks, which may result in a lack of functional connectivity between regions across subnetworks. Eigen-based decomposition pooling, as used in EigenGCN [18], causes a loss of crucial information about features due to the presence of multiple pooling layers. In our model, we considered both the structural and functional connectivity of brain data and used both to construct a graph for a subject, enhancing the classification accuracy. Table 2 highlights the findings of similar recent models and compares them with the proposed model.

## 4. Experimental Results

We used the Ubuntu operating system, an Intel i9 12900 processor, 16 GB RAM, and an Nvidia RTX A2000 GPU for all experiments conducted.

### 4.1. Dataset

The ABIDE is a preprocessed dataset used for autism brain imaging research. The dataset includes data from 16 sites, consisting of sMRI, rs-fMRI, and phenotypic information from 539 autistic patients and 573 typical controls. The data were harmonized and standardized to ensure consistent representation across all sites. The specific preprocessing steps performed on the dataset included the following:Motion correction: To reduce the impact of head movement on the fMRI data, the images were corrected for motion using tools such as MCFLIRT and MotionCorr.Spatial normalisation: To ensure a consistent representation of brain structures across participants, the data were transformed into a shared space using a standard brain template ( Montreal Neurological Institute template).Noise reduction: Various noise sources, such as physiological noise from the heart and respiratory system, were removed from the data using physiological regression or CompCor.Data cleaning: The data were checked for quality and outliers, and any poorly performing time points or participants were removed from the dataset.

Functional and structural preprocessing and the calculation of cortical measures were carried out using well-defined pipelines.

### 4.2. Parameter Search

The hyperparameters used for CNN models include the number of convolutional layers, the number of filters, the dropout rate, and the number of dense nodes in a dense layer. Various values that were used are mentioned in Table 3.

With the above hyperparameter values for CNN, a better accuracy score of 69.45 (Figure 7) was obtained with a 3D CNN model (number of convolutional blocks after the first layers = 4, filters = 64, dropout rate = 0.5, number of nodes in a dense layer = 1024).

### 4.3. Experiments

The population graph of a graph convolution model is formed with the similarity of features extracted from sMRI radiomics, and its edges contain the characteristics (1024) of the 19 retrieved derivatives. To demonstrate the significance of each summary measure and the useful outcome produced by combining the summary measurements, we conducted our experiment individually for each brain summary. Despite the variety in the topographical locations of the data collected, our model produced better results. The majority of deep learning models that use fMRI to diagnose autism have higher local accuracy levels. Our GCN model offers improved accuracy for multisite data in ABIDE, because it creates an edge between nodes based on similarities in the sMRI data.

### 4.4. Evaluation

The accuracy, specificity and sensitivity are commonly used metrics to evaluate the performance of deep learning models and can help to determine their efficiency.

The accuracy measures the proportion of correct predictions made by the model. It is a useful metric to evaluate the model’s overall performance, but it may not be the best metric for imbalanced datasets.
(6)$$Accuracy=\frac{\left(TP+TN\right)}{\left(TP+TN+FP+FN\right)}\cdot 100$$The sensitivity (also called the recall or true positive rate) measures the proportion of actual positive cases correctly identified by the model. It helps to detect false negatives and is essential in applications where the identification of positive cases is critical.
(7)$$Sensitivity=\frac{\left(fP+FN\right)}{\left(TP+TN+FP+FN\right)}\cdot 100$$The specificity (also called the true negative rate) measures the proportion of actual negative cases correctly identified by the model. It helps to detect false positives and is essential in applications where avoiding false alarms is critical.
(8)$$Specificity=\frac{TP}{\left(TP+FP\right)}\cdot 100$$The precision measures the accuracy of positive predictions made by the model. It tells us how many of the positive predictions made by the model were actually correct.
(9)$$Precision=\frac{TN}{\left(FP+TN\right)}\cdot 100$$

By examining the values of these metrics, we can gain insight into the strengths and weaknesses of the model’s performance (refer Figure 8). The standard deviations of these metrics for the proposed model combining 19 derivatives were **0.42%,0.44%,0.13%,** and **0.48%**, respectively.

## 5. Discussion

Most deep learning models use fMRI images to classify ASD patients and controls. The integration of structural and functional features provides a whole-brain analysis for better feature identification. Constructing a graph model based on the relationships among the sMRI features obtained using radiomics provides better relationships between individuals than using phenotypic data, as used in previous works. We performed experiments by using and ensembling the summary measures. Combining features extracted from various brain summaries provides a better classification result than applying a single summary measure. We integrated data from all sites and topographical locations in the ABIDE repository to ensure that the results were independent of demographic changes. Our model produced a better accuracy instead of leaving out a site strategy. The use of a graph convolutional model was shown to provide better results by grouping nodes based on their structural similarities.

### Comparison with the State-of-the-Art Methods

To demonstrate the overall performance of our brain network in the diagnosis of ASD, we compared our model with various state-of-the-art techniques. In addition, we compared our model with nongraph and GCN-based models to provide a comprehensive study.

To extract features and improve the classification performance, ASD-DiagNet uses a single-layer perceptron (SLP) and an autoencoder that are jointly learned. In contrast, when using the Restricted Boltzmann Machine, DBN features are learned from rs-fMRI and sMRI data (grey matter and white matter).

Based on the phenotypic data from the ABIDE dataset, the Siamese GCN (sGCN) learns patterns related to the similarities between graph nodes. In contrast, the EigenGCN operates on the eigenvectors of the adjacency matrix, which are used to encode the graph’s structure and related information. This makes it possible to run graph convolutions directly, rather than using the original graph representation.

A comparison of the efficiency of our model compared with various state-of-the-art models is presented in Table 4. ASD DiagNet uses Pearson’s correlation measure to determine the functional connectivity from the functional MRI data. Nineteen thousand, nine hundred features were selected, which was insufficient for training a deep learning model, so the Synthetic Minority Over-sampling Technique (SMOTE) was used for augmentation purposes. In our method, we merged the sMRI radiomic features with features extracted from the 3D CNN (48,544 + 1024 = 49,568) as the input for the autoencoders. The performance of the model was improved by 8.34%. Similarly, the merged features were fed to the DBN model, which uses a hierarchically restricted Boltzmann machine for feature learning. In this method, the average signals of each region of interest are calculated using the AAL atlas (116 ROI) and the grey and white matter volumes of the sMRI data. Our feature vector outperformed the model by producing an accuracy of 82.45% with a depth of 3. Considering graph-based networks, our model was proven to produce better results than the s-GCN and EigenGCN. The use of structural similarities to construct the edges of a graph was shown to be more efficient than using phenotypic data or graph kernels.

## 6. Conclusions

We designed and implemented a graph convolutional model to classify ASD using structural and rs-fMRI data information. The graphical model provides a better association between subjects, which helps us to classify them better. It is evident that fusing structural and rs-fMRI data helps to generate a well-learned graph and improves the classification accuracy. Our model is based on the entire brain’s functional connectivity and the brain’s grey matter, structure and shape. This distinguishes the behavioural, linguistic and social characteristics of patients with ASD from healthy controls. We determined the similarity between patients based on the structures of their brains, which proved to be a better similarity measure than phenotypical data, such as age, gender and location. Every model has some limitations. In our model, these are as follows: (i) the number of steps processed is higher when compared to those of other GCN models (EigenGCN and S-GCN); (ii) it takes more time to form the graph due to the formation of edges from the sMRI data; and (iii) the model is more complex due to the use of both sMRI and rs-fMRI data. Moreover, we used only the ABIDE dataset, and with a little pre-processing, our model could be tested on other MRI datasets such as the NDAR, open fMRI, ADNI, etc. Future research should consider the fusion of the INCLEN diagnostic dataset and videos captured during psychologists’ visits and MRI images of ASD patients to produce a better classification accuracy.

## Figures and Tables

**Figure 1 diagnostics-13-01143-f001:**
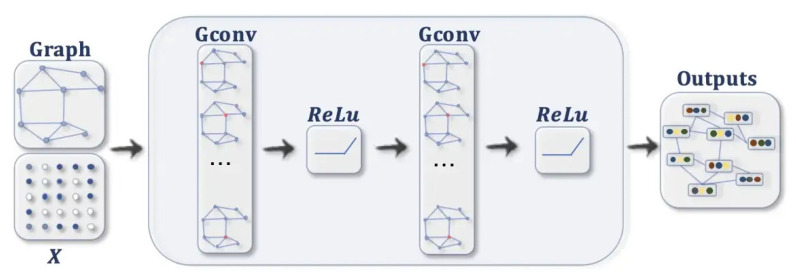
GCN Architecture.

**Figure 2 diagnostics-13-01143-f002:**
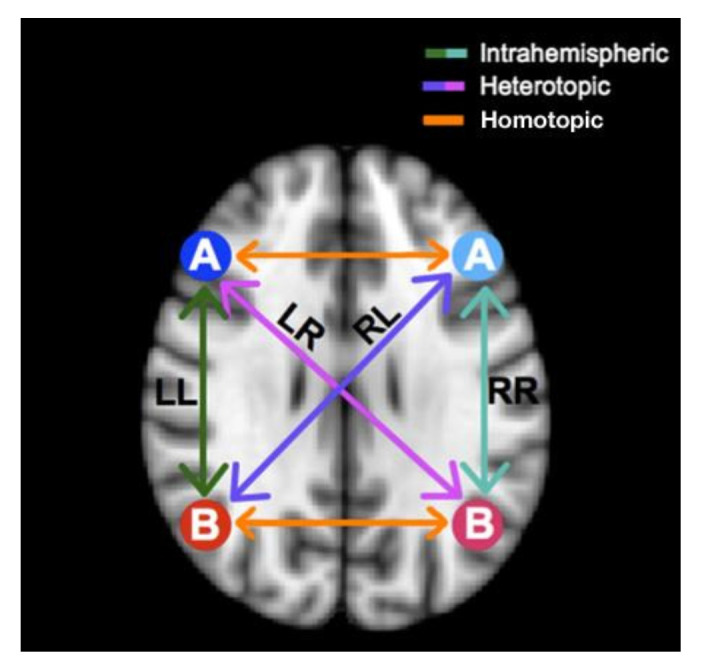
Comparison of Mirrored Voxels across Hemispheres in a Symmetrical Brain using VMHC.

**Figure 3 diagnostics-13-01143-f003:**
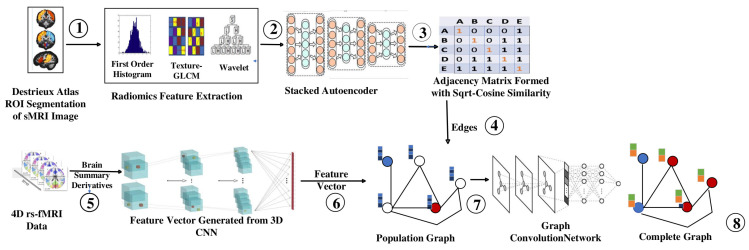
Workflow of Proposed Model. (1: ROI Extraction, 2: Features through Fischer’s Score, 3: Features through Stacked Autoencoder, 4: Builing Adjacency Matrix, 5: Extraction of Summary Measures, 6: Building Feature Vector for Each Node, 7: Constructing Population Graph, 8: Complete Graph using GCN).

**Figure 4 diagnostics-13-01143-f004:**
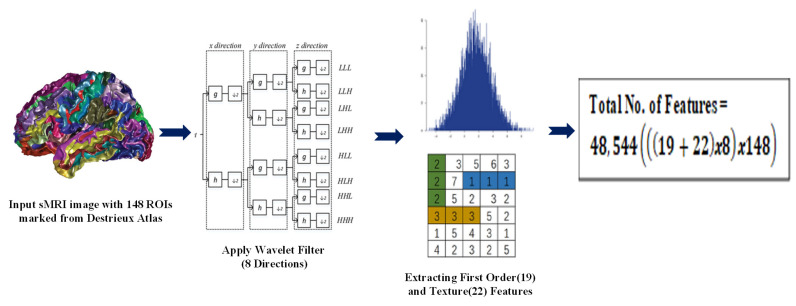
Radiomics Feature Extraction.

**Figure 5 diagnostics-13-01143-f005:**
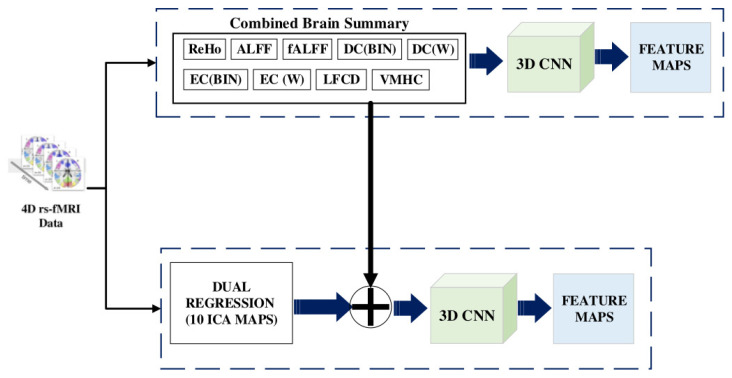
Feature Selection from rs-fMRI images.

**Figure 6 diagnostics-13-01143-f006:**
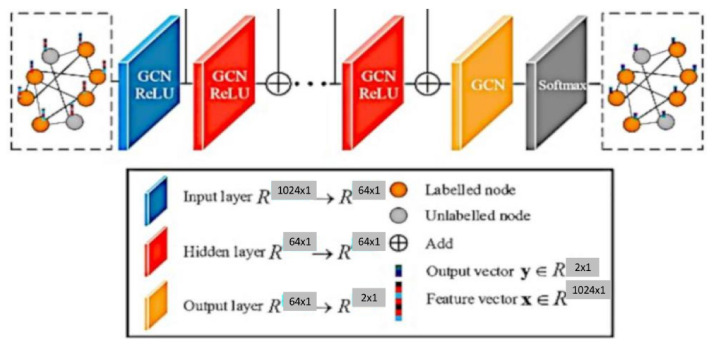
Graph Convolution.

**Figure 7 diagnostics-13-01143-f007:**
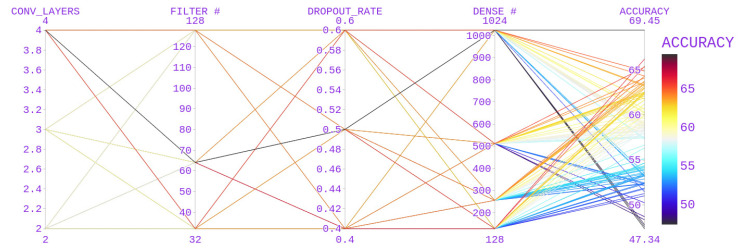
Parameter Selection for the 3D CNN.

**Figure 8 diagnostics-13-01143-f008:**
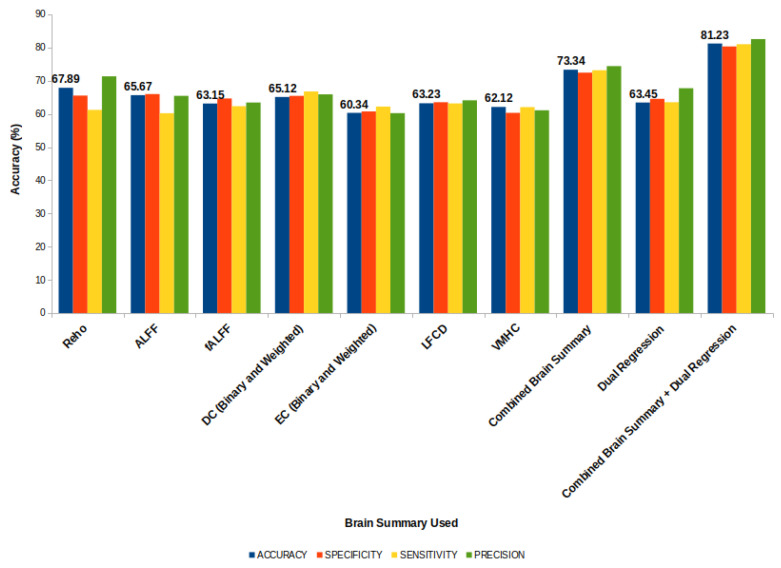
Accuracy of the Proposed Model.

**Table 1 diagnostics-13-01143-t001:** Summary of Related works.

Authors, Paper Title (Year)	Contribution	Drawback	Deep Learning Method	Dataset
MS Ahammed et al., Classification of ASD on Functional MRI Using Deep Neural Network (2021) [11]	Learned features of 3D fMRI images by increasing the number of convolution layers.	Topographical heterogeneity of dataset not considered	2D CNN	ABIDE1 (NYU)
MJ Leming et al., Single-participant structural similarity matrices lead to greater accuracy in classification of participants than function in autism in MRI (2021) [14]	Derived a structural similarity metric with grey matter volume data from structural MRI images	Generalisation of data not proven	2D CNN	Open fMRI, UK Biobank, ABIDE I, ABIDE II, NDAR
R Haweel et al., A CNN Deep Local and Global ASD Classification Approach with Continuous Wavelet Transform Using Task-Based FMRI (2021) [12]	Used K-means clustering to cluster BOLD signals in fMRI data and continuous wavelet transform to obtain a detailed representation of BOLD signals.	The choice of wavelet parameters is challenging and may affect the model’s performance	Clustered BOLD signals passed to CNN	NDAR
X Li et al., Two-Channel Convolutional 3D Deep Neural Network (2CC3D) for fMRI Analysis: ASD Classification and Feature Learning (2018) [13]	Two channels, the mean and standard deviation of temporal fMRI data are passed to the CNN.	The sliding window involves computing the convolution operation multiple times for overlapping regions of the input image. This can result in redundant computation, as the same features may be detected multiple times.	2 -Channel 3D CNN	Self
Eslami et al., A Hybrid Learning Approach for Detection of Autism Spectrum Disorder Using fMRI Data (2019) [15]	Used autoencoders to capture the complex patterns in fMRI data and single layer perceptron to provide a solution.	Used only the functional connectivity matrix features for the classification	Autoencoder and a Single–Layer Perceptron	ABIDE-I
H Sewani et al., An Autoencoder-Based Deep Learning Classifier for Efficient Diagnosis of Autism (2020) [16]	Autoencoder-based deep learning model followed by a sequence of two 1D CNNs for better feature learning.	Used only Pearson’s correlation matrix for extracting features from fMRI. No specific atlases or statistical measures were used to obtain regions of interest.	Autoencoder + 1D CNN	ABIDE
L Ali et.al, MMDD-Ensemble: A Multimodal Data–Driven Ensemble Approach for Parkinson’s Disease Detection (2021) [17]	Ensembling method for combining the features obtained from different modalities.	Voting after classification results does not ensure the feature selection accuracy of a modality used.	Machine Learning	Self
Y Ma et.al, Graph Convolutional Networks with Eigen Pooling (2019) [18]	Used Eigen Pooling to reduce the graph.	Eigen decomposition may lead to loss of important characteristics from a node	GCN	ENZYMES, PROTEINS, Mutagenicity
G Wen et.al, MVS-GCN: A prior brain structure learning-guided multi-view graph convolution network for autism spectrum disorder diagnosis (2022) [19]	Used graph structure learning to learn features from multiple views of a graph.	Used only fMRI and did not include the structural components of the brain	Graph Convolutional Network	ABIDE
X Song et.al, Graph convolution network with similarity awareness and adaptive calibration for disease-induced deterioration prediction (2021) [20]	Used the similarity between nodes for predictions and adaptive calibration for parameter tuning.	Used only fMRI and did not include the structural components of the brain	GCN	ADNI
Baiying Lei et.al, Multiscale enhanced graph convolutional network for mild cognitive impairment detection (2023) [21]	Used the functional connectivity matrix to construct a brain graph.	Structural features were not considered	Attention Graph Convolution Network	Self
S Parisot et.al, Disease prediction using graph convolutional networks: Application to Autism Spectrum Disorder and Alzheimer’s disease (2018) [22]	Used an autoencoder to obtain the features of a node from a graph and phenotypic data to build edges.	Using phenotypic data in a topographically heterogenous dataset to find similarities between nodes might reduce the model’s performance.	GCN	ABIDE
D Shi et al., Machine Learning for Detecting Parkinson’s Disease by Resting-State Functional Magnetic Resonance Imaging: A Multicentre Radiomics Analysis (2022) [23]	Used radiomics features to identify spontaneously abnormal brain activities as biomarkers.	Did not include cerebellum or multimodal data	Radiomics + SVM	figshare
L Wang et al., Textural features reflecting local activity of the hippocampus improve the diagnosis of Alzheimer’s disease and amnestic mild cognitive impairment: A radiomics study based on functional magnetic resonance imaging (2022) [24]	Investigated the use of textural features derived from functional magnetic resonance imaging (fMRI) of the hippocampus to improve the diagnosis.	Aimed to use imaging data to extract textural features that reflect local activity only in this region	Radiomics + Statistics	Self

**Table 2 diagnostics-13-01143-t002:** Comparison of Proposed Model with Recent Similar Models.

S.No	Model	Methodology	Findings
1	ASD-DiagNet	Feature Map obtained from VAE [rs-fMRI]	Correlation coefficients can be influenced by head motion, the signal-to-noise ratio and individual differences in brain anatomy
2	DBN	Feature Map obtained from the Restricted Boltzmann Machine [rs-fMRI]	Correlation coefficients can be influenced by head motion, the signal-to-noise ratio and individual differences in brain anatomy
3	MMDC	Classification results of different modalities (phonation: voicing of vowels, voiced and unvoiced) are ensembled by blending or voting	Ensembling multiple modalities before classification can improve the accuracy compared to the ensembling of classification results
4	s-GCN	Nodes-> Atlas [rs-fMRI] Edges-> Phenotype data	The use of phenotype data for constructing edges might lead to a lack of heterogeneity since phenotype data do not consider variation in topographical locations
5	EigenGCN	Nodes-> Functional Connectivity Matrix (FCM) [rs-fMRI] Edges-> Graph Kernel Function [FCM]	Loss of crucial information about features due to multiple pooling layers
6	MVS-GCN	Brain graph constructed with multiple views from the subnetwork	The heterogeneity of the brain is not considered when forming subnetworks, which may result in a lack of functional connectivity between regions across subnetworks.
**7**	**Proposed Model**	**Nodes-> Feature Maps obtained from Combined BS + Dual Regression** **Edges->Similarity Measure calculated using Radiomics from sMRI**	-

**Table 3 diagnostics-13-01143-t003:** Hyperparameter Selection.

Hyper Parameters	Values
No. of Convolutional blocks after the First layer	2,3,**4**
Number of Filters	32,**64**,128
Dropout Rate	0.4, **0.5**, 0.6
No. of nodes in the dense layer	128,256,512,**1024**

**Table 4 diagnostics-13-01143-t004:** Comparison of the Proposed Model with State-of Art Models.

S.No	Model	Accuracy	Sensitivity	Specificity
1	ASD-DiagNet	70.3	68.3	72.2
2	DBN	65.56	84	32.96
3	s-GCN	65.74	64.73	60.12
4	EigenGCN	57.5	58.81	59.94
5	MVS-GCN	69.38	69.93	71.22
**6**	**Proposed Model**	**81.23**	**81.36**	**81.02**

## Data Availability

The data used in this study is from public database ABIDE. The data is available at http://fcon_1000.projects.nitrc.org/indi/abide/.

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
