# Peer review of "A Model for Diagnosing Autism Patients Using Spatial and Statistical Measures Using rs-fMRI and sMRI by Adopting Graphical Neural Networks"

_diagnostics, 2023, doi:10.3390/diagnostics13061143_

Round 1
Reviewer 1 Report
Authors developed a graph convolutional model to classify ASD from Typical Controls. However, the paper seems lack of innovations.
1. It seems that the two proposed methods in Section 3.2 have no obvious differences except they use two different inputs. The 3D CNN has no special features in which the convolution and the max polling are repeated by five times. Authors are recommended to describe how the derivatives are combined together before they are input to the 3D CNN more clearly, and to introduce the reasons why the 3D convolutional model adopts such a structure.
2. The experiments are expected to be implemented further. It is not enough that the proposed methods are compared with the chosen brain summary measures which are not new. Comparisons with the state-of-the-art methods are necessary. Figure 10 to Figure 13 are unnecessary since the accuracies are shown in Tab.4 completely.
3.Ref[17] probably was cited by mistake. Its contents are not consistent with the description in Line 117-120.
4.Fig.4 illustrates the 1-D wavelet package transform was used. However, in the paper the 2-D wavelet transform or 2-D wavelet package transform should be used to decompose the ROIs as the input since each ROI is still an image. Therefore, wavelet-based features in more directions will be obtained instead of 8 directions after the 2-layer transform.
Reviewer 2 Report
The manuscript sounds technically poor, I have following concerns should be addressed before any decision.
*The existing literature should be classified and systematically reviewed, instead of being independently introduced one-by-one.
*The abstract is too general and not prepared objectively. It should briefly highlight the paper's novelty as what is the main problem, how has it been resolved and where the novelty lies?
*The 'conclusions' are a key component of the paper. It should complement the 'abstract' and normally used by experts to value the paper's engineering content. In general, it should sum up the most important outcomes of the paper. It should simply provide critical facts and figures achieved in this paper for supporting the claims.
*For better readability, the authors may expand the abbreviations at every first occurrence.
*The author should provide only relevant information related to this paper and reserve more space for the proposed framework.
*However, the author should compare the proposed algorithm with other recent works or provide a discussion. Otherwise, it's hard for the reader to identify the novelty and contribution of this work.
*The descriptions given in this proposed scheme are not sufficient that this manuscript only adopted a variety of existing methods to complete the experiment where there are no strong hypothesis and methodical theoretical arguments. Therefore, the reviewer considers that this paper needs more works.
*Key contribution and novelty has not been detailed in manuscript. Please include it in the introduction section
*What are the limitations of the related works
*Are there any limitations of this carried out study?
*How to select and optimize the user-defined parameters in the proposed model?
*There are quite a few abbreviations are used in the manuscript. It is suggested to use a table to host all the frequently used abbreviations with their descriptions to improve the readability
*Explain the evaluation metrics and justify why those evaluation metrics are used?
*Some sentences are too long to follow; it is suggested that to break them down into short but meaningful ones to make the manuscript readable.
*The title is pretty deceptive and does not address the problem completely.
*Every time a method/formula is used for something, it needs to be justified by either (a) prior work showing the superiority of this method, or (b) by your experiments showing its advantage over prior work methods - comparison is needed, or (c) formal proof of optimality. Please consider more prior works.
*The data is not described. Proper data description should contain the number of data items, number of parameters, distribution analysis of parameters, and of the target parameter itself for classification.
* The related works section is very short and no benefits from it. I suggest increasing the number of studies and add a new discussion there to show the advantage. Following studies can be used.
a. Automated detection of brain abnormality using deep-learning-scheme: A study
b. MMDD-Ensemble: A Multimodal Data–Driven Ensemble Approach for Parkinson's Disease Detection
c. A unified design of ACO and skewness based brain tumor segmentation and classification from MRI scans
d. Review of Automated Computerized Methods for Brain Tumor Segmentation and Classification
*Use Anova test to record the significant difference between performance of the proposed and existing methods.
Reviewer 3 Report
The article entitled “ Graphical Neural Networks for Diagnosing Autism Patients using Spatial and Statistical Measures with rs-fMRI and sMRI” is well-written and, from my point of view, would be of interest to the readers of MDPI Diagnostics. However, despite these, and before its publication, I would recommend the following changes:
Materials and methods: A more detailed technical explanation of the proposed method is required, and it must be improved. Authors must cite previous works and discuss the well-motivated differences with the proposed work. It is noteworthy that similar models were proposed between 2017 and 2022. The authors must choose the latest models as fair baselines. So, the bibliography employed for the materials and method must be extended and improved.
The results are very well presented, but the sample of 61 slices of 61X73 images is not remarkable. Also, Figures 3, 4, and 5 should be clarified and must be explained in more detail. Also, it is almost impossible for another researcher to repeat the results following the proposed method.
The discussion section needs to be sufficiently discussed. The weak points of this study should be listed, and future improvement methods must be described.
Round 2
Reviewer 1 Report
1 Although the authors revised the article according to the review comments, the subscript is hard to read since it mixes the corrections with revisions. Authors are recommended to submit the latest version.
2 In Page 3, the developed brain summary measures are recommended to be introduced briefly since readers can learn them from the literatures.
Reviewer 2 Report
The paper is well revised can be accepted.
Author Response
Respected Sir,
Thank you for your support and constructive remarks on our paper. Your comments helped in reconstructing the paper more efficiently
Again, thank you for your support and enlightening comments on our paper.
Reviewer 3 Report
The authors did not correct the article according to the suggestions given in the previous version.
Round 3
Reviewer 1 Report
1 Algorithms 1, 2 and 3 are recommended to be omitted since they only show how to perform the calculation and have no prominences. Moreover, Tables 2 to 4 can be placed in the appendix.
2 Figure 8 and Table 7 show the same results. Therefore, it is recommended to keep only one and delete another. Similarly, Figure 9 is not necessary.
3 The introduction of State-of-the-art models in Section 5.1.1 and 5.1.2 seems too long-winded. They are recommended to be introduced briefly.
Author Response
Respected Sir,
Please see the attachment

Reviewer 3 Report
Accept the manuscript in its current form.
Author Response
Respected Sir,
Your comments helped in improving the "Materials and Methods " section. Thank you for providing constructive comments, which helped in improving our manuscript